# Data Mixture Optimization:
# A Multi-fidelity Multi-scale Bayesian Framework

**Thomson Yen**
Decision, Risk, and Operations Division
Columbia Business School
ty2531@columbia.edu

**Andrew Wei Tung Siah**
Decision, Risk, and Operations Division
Columbia Business School
andrew.siah@columbia.edu

**Haozhe Chen**
Decision, Risk, and Operations Division
Columbia Business School
haozhe.chen@columbia.edu

**Tianyi Peng**
Decision, Risk, and Operations Division
Columbia Business School
tp2845@columbia.edu

**Daniel Guetta**
Decision, Risk, and Operations Division
Columbia Business School
crg2133@columbia.edu

**Hongseok Namkoong**
Decision, Risk, and Operations Division
Columbia Business School
hongseok.namkoong@columbia.edu

## Abstract

Careful curation of data sources can significantly improve the performance of LLM pre-training, but predominant approaches rely heavily on intuition or costly trial-and-error, making them difficult to generalize across different data domains and downstream tasks. Although scaling laws can provide a principled and general approach for data curation, standard deterministic extrapolation from small-scale experiments to larger scales requires strong assumptions on the reliability of such extrapolation, whose brittleness has been highlighted in prior works. In this paper, we introduce a *probabilistic extrapolation framework* for data mixture optimization that avoids rigid assumptions and explicitly models the uncertainty in performance across decision variables. We formulate data curation as a sequential decision-making problem—multi-fidelity, multi-scale Bayesian optimization—where {data mixtures, model scale, training steps} are adaptively selected to balance training cost and potential information gain. Our framework naturally gives rise to algorithm prototypes that leverage noisy information from inexpensive experiments to systematically inform costly training decisions. To accelerate methodological progress, we build a simulator based on 472 language model pre-training runs with varying data compositions from the SlimPajama dataset. We observe that even simple kernels and acquisition functions can enable principled decisions across training models from 20M to 1B parameters and achieve **2.6x** and **3.3x** speedups compared to multi-fidelity Bayesian optimization and random search baselines. Taken together, our framework underscores potential efficiency gains achievable by developing principled and transferable data mixture optimization methods. Our code is available at https://github.com/namkoong-lab/data-recipes.

## 1 Introduction

*Data is the foundational infrastructure upon which all AI systems are built.* Scaling data has been a key driver of progress in machine learning, particularly in language model training (Deng et al., 2009;

39th Conference on Neural Information Processing Systems (NeurIPS 2025).

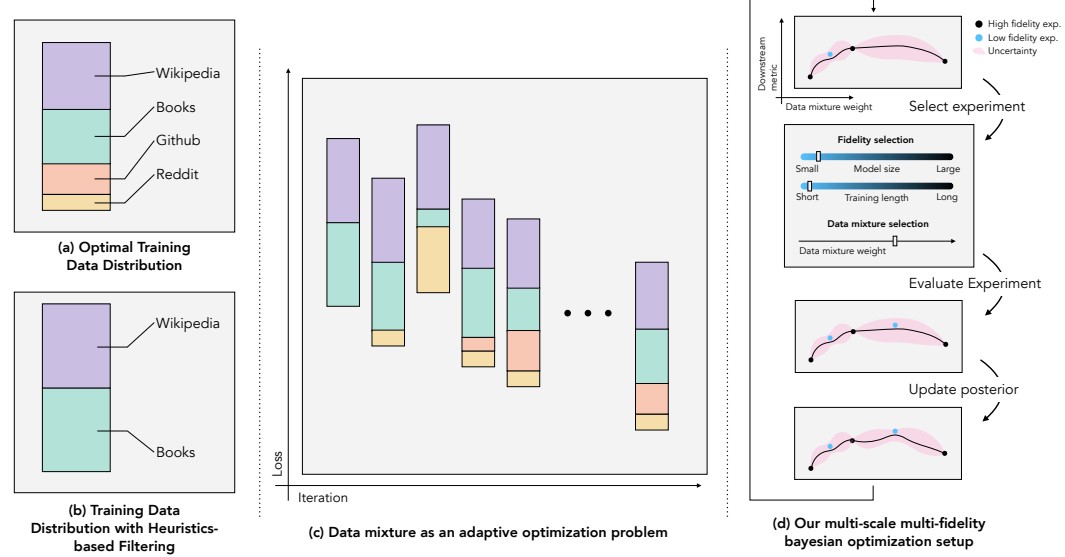

Figure 1: Our multi-fidelity multi-scale Bayesian optimization (BO) framework. (a) Given an unknown optimal training data distribution, (b) existing methods use heuristic-based filtering techniques to approximate the optimal distribution. (c) Our algorithm treats data mixture optimization as a BO problem. (d) We explore data mixtures in a cost-aware fashion; when we test a new data mixture, we also choose the *fidelity* of the observation we will observe. Larger models trained for more steps will result in *high fidelity* observations, but be more expensive. Every point we observe updates our probabilistic belief of model performance over the data mixture, model size, and training steps space, which guides subsequent parameters.

Hoffmann et al., 2022a; Gadre et al., 2024). While this data-centric approach has led to impressive performance gains, it also incurs substantial computational and financial costs in training state-of-the-art models (Hoffmann et al., 2022a; Luccioni et al., 2023). Beyond raw scale, the *composition of training data* has emerged as a critical factor in model performance (Albalak et al., 2023a; Goyal et al., 2024a). For instance, TinyStories (Eldan and Li, 2023) demonstrated that models with only 10 million parameters, when trained on a carefully chosen synthetic dataset, can generate coherent and consistent English text — surpassing the capabilities of significantly larger models like GPT-2 Small (125 million parameters) (Radford et al., 2019). Similarly, Li et al. (2024) showed that a carefully curated dataset enables training a 7-billion-parameter model comparable in performance to that of Mistral-7Bv0.3 and Llama 3 8B while using six times less compute. In practical scenarios where heterogeneous data sources are available, the choice of training mixture has been shown to significantly impact model performance.

**Pitfalls of Prior Approaches:** This growing recognition of data composition's importance has led institutions to develop proprietary data mixtures based on domain expertise and empirical observations (Radford et al., 2021; Jiang et al., 2023; OpenAI, 2024). Others have introduced heuristics, such as Wikipedia upsampling and perplexity-guided data selection, to refine training mixtures (Thrush et al., 2024; Blakeney et al., 2024). However, these ad hoc approaches are often tailored to specific training datasets and downstream tasks, and may *fail to transfer* across domains and data types. For instance, when organizations in specialized sectors such as healthcare or finance seek to train custom language models on proprietary datasets, it remains unclear whether heuristics developed for public datasets are still effective. Given the substantial resources required for training high-performance language models, there is a pressing need for a principled framework to address data mixture optimization.

Another line of research attempts to deterministically identify the functional relationship between data composition and model performance (Ye et al., 2024; Ge et al., 2025a; Liu et al., 2025). Data Mixing Laws (Ye et al., 2024), for example, proposes fitting validation losses as an exponential function of linear combinations of data proportions. However, collecting sufficient data to fit parameters of such functions at the desired scale and training duration is often computationally prohibitive. As a result, these methods inevitably rely on extrapolating the parameters learned from smaller models trained for fewer steps. Numerous studies have highlighted the *brittleness* of using such naive extrapolation

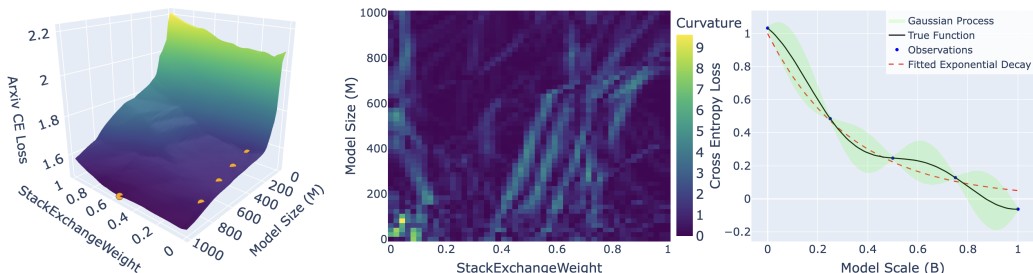

Figure 2: **Left:** The predicted validation cross-entropy loss on ArXiv data (Shen et al., 2024) as a function of data mixing coefficient and model sizes from a data-driven predictor on 472 runs (see details in Sec. 2). Notice the highly non-smooth geometry. Orange dots highlight the optimal data mixture proportion for each model scale. Note that they are not consistent across scales. **Middle:** The curvature at these points shows there are points of high irregularities, suggesting that the relationship between data mixture and model performance is unlikely to take a simple functional form. **Right:** A demonstration showing how functional forms like exponential decay fitted on a small number of points would result in a high predictive error. In contrast, a probabilistic model such as a Gaussian Process can capture uncertainty over the points.

to guide hyperparameter decisions (Levine et al., 2020; Yang et al., 2021; Jiang et al., 2025). Notably, Jiang et al. (2025) demonstrated that extrapolating validation losses based on small models often leads to *inaccurate* results. We reinforce this finding in our empirical study, observing that the optimal data mixing proportion does not remain constant across model scales (see Figure. 2)

Existing approaches, whether based on heuristic trial-and-error or deterministic but unverifiable extrapolation, risk failing to identify the optimal data mixture as experiment scale or data domains change. Moreover, they share another fundamental limitation: when additional computational budget becomes available for training new models, these approaches offer no clear guidance on *how to select the next promising data mixture to experiment with*. To address these challenges, we propose viewing the problem of curating the optimal data mixture as an *adaptive optimization* problem, where practitioners iteratively refine their mixing decisions based on empirical observations from prior experiments. This framework leverages the intuition that model performance exhibits local consistency across similar mixtures, training steps, and model scales, while avoiding rigid assumptions about the global structure of the performance landscape.

**Our Formulation:** Sequential optimization of data mixtures necessitates comprehending which data compositions suffer the highest uncertainty and sharpening beliefs on performance as more observations are gathered. In particular, good adaptive policies must distinguish between aleatoric and epistemic uncertainty: epistemic uncertainty can be reduced with more data, while aleatoric uncertainty is irreducible. Measurements must be planned to maximally reduce epistemic uncertainty on future runs by balancing exploration and exploitation.

We formulate this sequential optimization framework as a *Bayesian optimization* (BO) problem: we maintain probabilistic beliefs on the performance of various data mixtures and model scales, and use these beliefs to choose the next model scale to train, on what data mixture, and for how long. Once we fit and evaluate this new model, we use its performance to update our beliefs (Hutter et al., 2011; Falkner et al., 2018; Frazier, 2018).

In standard BO, the cost of each new observation is the same, and we aim to optimize an objective while observing the smallest number of points possible. Our setting presents *additional* challenges – the cost of training a new model and observing its performance is affected by (1) the number of steps for which the model is trained and (2) the scale of the model (the number of parameters therein).

The number of steps for which a model is trained affects the *quality* of the observation – the more steps we use to train the model, the more accurately the results will reflect the utility of training on the data mixture in question. Previous work has handled this conundrum using so-called *multi-fidelity* BO, in which evaluations are 'stopped early' during the training process if it becomes clear the information revealed during additional training steps will not be worth the expense (Swersky et al., 2014; Domhan et al., 2015; Kandasamy et al., 2017; Li et al., 2018a).

***Multi-Fidelity Multi-Scale*** (MFMS) **Bayesian Optimization:** Our setting is distinguished by the second factor above — we also want to use data gathered on smaller model scales to guide our search

for optimal parameters in larger models. We refer to this new setting as *Multi-Fidelity Multi-Scale* (MFMS) BO. While it can be viewed as a special case of the broader framework that optimizes black-box functions with access to cheaper approximate evaluations (Poloczek et al., 2016), the MFMS setting introduces a distinctive and interesting structure. Specifically, the model scales dimension differs fundamentally from the usual fidelity dimension associated with training steps. For one, the number of training steps typically far exceeds the number of model scales considered. More importantly, when training for $z$ steps, we naturally obtain observations for all intermediate steps up to $z$. In contrast, evaluating a model of size $m$ provides no inherent information about the performance of smaller or larger architectures. This raises interesting questions about how to appropriately treat and exploit this structure, opening new methodological directions for investigation.

Fortunately, unlike conventional hyperparameters such as learning rate or momentum, where optimal configurations exhibit complex scaling behavior across model sizes (Yang et al., 2021), recent empirical evidence suggests that optimal data mixture compositions enjoy *greater transferability* from smaller to larger model architectures (Ye et al., 2024; Ge et al., 2025a). This transferability property enables the strategic use of smaller-scale evaluations to identify optimal data mixture configurations that remain effective at target model scales, substantially reducing the computational cost of the optimization process.

The main contributions of the paper are as follows:

- We propose a *probabilistic extrapolation* framework to address the problem of optimizing data mixture for training LLMs. The framework avoids rigid assumptions on the functional dependence between decision variables and model performance by explicitly modeling the performance uncertainty, thereby highlighting the need to strategically experiment with selected data mixtures to minimize the uncertainty in the optimal data mixture at the desired scale.

- The framework gives rise to a *Multi-Fidelity Multi-Scale* BO problem, which provides a principled foundation for developing and evaluating transferable methods for optimizing data mixture. While the BO formalism that treats cheaper but noisy evaluations of the cost function of interest is well established, applying it to data mixture optimization introduces distinctive structural features, stemming from the interplays between training steps and model scales, that opens up interesting avenues for future methodological advancements, including batching strategies, asynchronous optimization, custom Gaussian process kernels, and look-ahead methods.

- To motivate the MFMS setting, we illustrate the use of smaller model sizes and earlier training steps for informing large-scale training decisions. We empirically show that exploiting cheaper evaluations on both dimensions enable more accurate predictions of model's performance at larger scale (Section 2).

- To spur methodological progress, we build an empirical testbed based on a simulator trained on 472 language model pre-training runs with varying data compositions from the SlimPajama dataset. We demonstrate the promises of the MFMS BO setting by introducing a Gaussian-process-based method. We compare the method against baselines such as Hyperband (Li et al., 2018a) and Random Search, which do not consider model scale as a decision variable. We find that even this simple approach can better explore different data mixtures and model scales, and deliver the best terminal model (as measured by downstream task performance) at least **2.6x** faster than baselines (Section 4).

## 2 Motivation for Multi-fidelity Multi-scale Framework

While deterministically extrapolating optimal data mixtures from small-scale experiments to large-scale models can be unreliable, intuitively, results from smaller models and earlier training steps still provide valuable information that can guide large-scale training decisions. In this section, we experimentally validate two key premises of MFMS BO: (1) smaller models can help predict the performance of larger models under various data mixtures, and (2) undertrained models trained for fewer steps over different data mixtures can inform the performance of fully-trained models.

To demonstrate the predictive utility of smaller-scale experiments, we train predictors that take language model training parameters — {data mixtures, model scale, training step} — as input and

| | Train | Test |
|---|---|---|
| $E_1$ | half of 1B runs | remaining 1B runs |
| $E_2$ | half of 1B runs + 700M runs | remaining 1B runs |
| $E_3$ | half of 1B runs + all smaller runs | remaining 1B runs |
| $E_4$ | half of 700M runs | remaining 700M runs |
| $E_5$ | half of 700M runs + 500M runs | remaining 700M runs |

Table 1: Model size experiments

| Dataset | $E_6$ | $E_7$ | $E_8$ |
|---|---|---|---|
| $R^2$ | 0.69 | 0.77 | **0.82** |
| $R^2(\log)$ | 0.74 | 0.82 | **0.85** |

Table 2: Results averaged over 3 runs. Notice the predictive power of our MLP is strongest when it is trained on many runs for fewer steps. This provides support for the intuition that one should distribute fixed compute budgets across multiple shorter training sessions rather than fewer longer ones.

predict either validation losses or downstream task accuracies of the resulting language model. Our experiments show that training the predictors on results from smaller-scale experiments improves the predictors' ability to estimate model performance as a function of data mixture at larger scales.

*It is important to note that these predictors differ from the language models. The predictors take training hyperparameters as input and forecast the resulting language model's performance.*

## 2.1 Collecting Predictor's Data Through LLM Pre-training Runs

To generate the training data for our predictors, we pretrained 472 language models using the OLMo 2 package (OLMo et al., 2024) with datasets derived from SlimPajama (Shen et al., 2024), a deduplicated subset of RedPajama (Weber et al., 2024). We used five categories of SlimPajama data for pretraining: *Wikipedia*, *StackExchange*, *Github*, *ArXiv*, and *Book*. Data from the *CommonCrawl* and *C4* categories were held out to simulate out-of-distribution scenarios. The proportions of these five categories in each run were randomly sampled from a Dirichlet distribution. We pretrained language models varying in size from 20M to 1B parameters. We recorded training losses, validation losses across all seven data categories, and evaluated performance on three downstream tasks: *HellaSwag*, *PIQA*, and *Arc Easy* (Zellers et al., 2019; Bisk et al., 2020; Clark et al., 2018). The entire dataset was collected using 4x NVIDIA H100 80GB HBM3 for **500** compute days. Additional implementation details for pretraining are provided in Appendix A.

## 2.2 Predictor Training

We trained predictors using multilayer perceptrons (MLPs) that take as input the model size, number of training steps, and dataset proportions from the five pretraining categories. These predictors output predictions for training loss, validation losses across seven categories (including the held-out *CommonCrawl* and *C4*), and downstream task accuracies for *HellaSwag*, *PIQA*, and *Arc Easy*. Predictor performance was measured using the coefficient of determination ($R^2$). Complete training details are documented in Appendix B.

## 2.3 Small Models Help Predict Larger Models Outcomes

We begin by investigating the extent to which smaller model runs can inform the dynamics of larger ones, as the literature on scaling laws (Gadre et al., 2024) would suggest. Table 1 details these experiments, and Table 3 lists the results of the experiment.

We note that, as expected, information garnered from training runs on *smaller* models seems to considerably increase the accuracy of our predictions on *larger* models, motivating our hope that a carefully crafted optimization algorithm can exploit the relationship.

Unsurprisingly, we note that $E_3 \approx E_2 > E_1$ and $E_5 > E_4$: the closer in scale the smaller models are to the larger model about which we wish to make a prediction, the more useful the information is. We, therefore, expect our MFMS BO algorithm to 'step through' model scales, starting with small and cheap models to identify promising data mixtures, and then progressing to larger models, all the while refining the data mixtures it considers optimal.

## 2.4 Earlier Training Steps Help Predict Later Training Steps

| Dataset | $E_1$ | $E_2$ | $E_3$ | $E_4$ | $E_5$ |
|---|---|---|---|---|---|
| Wikipedia | 0.75 | **0.96** | 0.94 | 0.73 | **0.88** |
| ArXiv | 0.68 | **0.92** | 0.93 | 0.59 | **0.82** |
| Github | 0.66 | 0.95 | **0.95** | 0.62 | **0.87** |
| Book | 0.83 | **0.97** | 0.97 | 0.79 | **0.92** |
| StackExchange | 0.73 | **0.95** | 0.95 | 0.68 | **0.90** |
| CommonCrawl | 0.84 | **0.98** | 0.98 | 0.81 | **0.94** |
| C4 | 0.86 | **0.99** | 0.98 | 0.82 | **0.95** |
| ArcEasy | 0.92 | **0.94** | 0.94 | 0.88 | **0.90** |
| HellaSwag | 0.97 | **0.98** | 0.97 | 0.94 | **0.96** |
| PIQA | 0.94 | **0.96** | 0.96 | 0.90 | **0.93** |

Table 3: $R^2$ values of the experiments listed in Table 1, averaged over 3 random seeds. Notice that $E_2 > E_1$ and $E_5 > E_4$ – our ability to predict the performance of larger models is considerably enhanced by insights from smaller models. Note also that $E_3 \approx E_2$; adding information about *much* smaller models does not seem to help.

The second central premise of our approach is that, given a fixed compute budget, it can be better to attempt many runs for fewer training steps than fewer runs for a larger number of training steps. To test this hypothesis, we carried out three additional experiments. In each of these experiments, we attempt to predict the final losses in 30% of our model runs (evenly distributed across model sizes). The MLP for each of these experiments is trained on (1) a set of complete runs, one for each model size (2) a set of 'truncated' runs, evenly distributed across model sizes. In $E_6$, we use 16 runs truncated at 19600 training steps, in $E_7$, we use 22 runs truncated at 13000 training steps, and in $E_8$, we use 32 runs truncated at 8500 steps; thus, these experiments are trained on numbers generated with the *same* FLOPS budget. As expected, $E_8$ results in the most performant MLP model, providing evidence for our second central premise.

## 3 Multi-Fidelity Multi-Scale Bayesian Optimization

Knowing that both smaller models and early stopping can provide valuable insights for optimizing data mixtures, practitioners face a critical dilemma: given a fixed computational budget, how should one allocate resources to train the best model? Should one train many small models to explore different data mixtures, use early stopping at the target scale to abandon poor-performing data mixtures, or adopt a hybrid approach combining both strategies? The MFMS setting articulates this dilemma, and in this section, we provide a mathematical formulation of the problem.

We consider having access to a set of $n$ datasets $\mathcal{D} = \{D_1, D_2, \ldots, D_n\}$, and aim to train the best-performing language model, evaluated on a given metric, with $m^*$ parameters for $z^*$ training steps using $T$ datapoints. The key decision variables are the fractions of the data budget $T$ allocated to each dataset. Specifically, we sample $w_i T$ data points from $D_i$, where $\boldsymbol{w} = \{w_1, w_2, ..., w_n\} \in \Delta^n$ and $\Delta^n$ denotes $n$-dimensional probability simplex. Let $\mu(\boldsymbol{w}, m, z)$ denote the performance of a model with $m$ parameters trained for $z$ training steps using dataset proportions $\boldsymbol{w}$.

The model's performance is represented by an unknown function $\mu(\boldsymbol{w}, m^*, z^*)$, which maps a given data mixture $\boldsymbol{w}$, model size $m$, and training steps $z$ to an evaluation metric. Our goal is to solve the optimization problem, $\arg\max_{\boldsymbol{w}} \mu(\boldsymbol{w}, m^*, z^*)$, where $m^*$ and $z^*$ represent the target model size and training steps.

We have a budget $B$ with which we can experiment using different values of $\boldsymbol{w}$, $m$, and $z$. Each evaluation of $\mu(\boldsymbol{w}, m, z)$ incurs a cost $c(m, z)$. $c(m, z)$ is typically an increasing function of $m$ and $z$, though our framework does not require this. Using $m^*$ parameters and $z^*$ steps every time we evaluate a new set of weights would quickly exhaust our budget. Instead, therefore, we might probe a particular set of weights on a smaller model with $m < m^*$ parameters, or with $z < z^*$ training steps — while the resulting observation $\mu(\boldsymbol{w}, m, z)$ would be less informative than $\mu(\boldsymbol{w}, m^*, z^*)$, it would be considerably cheaper and still provide valuable information.

However, since $\mu(\cdot)$ is an unknown function, it is critical to address the uncertainty in its values. BO provides a natural framework that explicitly models uncertainty in $\mu(\cdot)$ and systematically refines estimates of $\mu(\cdot)$ through posterior updates, as one observes more evaluations at different input configurations. This technique is called multi-fidelity BO. See Figure 1 for a graphical representation of the setup.

Traditional approaches to fidelity-aware BO primarily address scenarios in which the model architecture $m$ remains fixed and only the number of training steps $z$ is varied (Swersky et al., 2014; Domhan et al., 2015; Kandasamy et al., 2017; Li et al., 2018a). We add a layer of complexity by also considering model scale. One might be tempted to regard model scale as merely an additional fidelity

dimension, otherwise identical to training steps. However, we note that there is a *fundamental distinction* between the two dimensions: In the course of evaluating a model trained for $z$ training steps, we must also evaluate that model for all steps $z' < z$, whereas no such hierarchical relationship exists for evaluations across different model scales. This structural difference prevents a direct application of the prior techniques, and suggests promising avenues for novel methodological developments in multi-fidelity optimization theory, though such extensions lie beyond the scope of our present work.

## 4 Experimental Setup

### 4.1 Evaluation

We propose MFMS BO as a natural framework to find optimal data mixture in practice. To demonstrate the potential of our framework, we adapt a BO algorithm based on Gaussian processes (GPs) to optimize the data mixture. However, training a new language model for each parameter that the algorithm proposes would be prohibitively expensive. Instead, therefore, we train a predictor as described in Section 2.2, and use the predictor to benchmark our approach. We train the predictor on 472 language model training runs described in Section 2.1, where 422 of the runs are randomly selected as a training set and the remaining 50 runs as a validation set. The predictor achieves $R^2 > 0.95$ across all metrics, suggesting that it is suitable as a surrogate benchmark (Eggensperger et al., 2022; Pfisterer et al., 2022).

*Note: the predictor is not necessary for applying our algorithm in real-world scenarios. It serves only as an efficient benchmarking tool.*

### 4.2 Multi-fidelity Multi-scale Gaussian Process (MFMS-GP)

---

**Algorithm 1** Gaussian Process and EIpu

---
1: **Input:** Probability space $\Delta^n$, model-scale space $\mathcal{M}$, training-step space $\mathcal{Z}$, and cost function $c(\cdot, \cdot)$
2: Initialize Gaussian Process (GP) surrogate model with three RBF kernels over $\Delta^n$, $\mathcal{M}$, and $\mathcal{Z}$ and a linear mean function
3: Randomly sample points from $\Delta^n$, $\mathcal{M}$, and $\mathcal{Z}$ to initialize hyperparameters of GP.
4: Initialize history $\mathcal{H}$ with the randomly sampled points
5: **for** each optimization iteration **do**
6:     **for** each $(m, z) \in \mathcal{M} \times \mathcal{Z}$ **do**
7:         Optimize the EI evaluated at these values of $m$ and $z$ with respect to $\boldsymbol{w}$ gradient descent
8:     **end for**
9:     Select the next configuration $\lambda_{\text{next}} = (\boldsymbol{w}_{\text{next}}, m_{\text{next}}, z_{\text{next}})$ that maximizees the Expected Improvement per Unit (EIpu):

$$\text{EIpu}(\lambda) = \frac{\text{EI}(\lambda)}{c(m, z)}$$

10:     Evaluate $\mu(\lambda_{\text{next}})$
11:     Store results in $\mathcal{H}$
12:     Update posterior of GP with $\mathcal{H}$
13: **end for**
14: Return best $\boldsymbol{w}^* = \lambda^*[0]$ from configuration $\lambda^* = \arg\max_{\lambda \in \mathcal{H}} \mu(\lambda)$

---

We implement a GP surrogate model for our MFMS setting. The kernel of the GP is a product of three separate RBF kernels for the data proportion, the model scale, and the training step dimensions. To enable learning the positive correlation between model performance and both model scales and training steps, we use a linear mean function.

For the acquisition function, we use Expected Improvement (EI). EI aims to quantify the expected gain over the current best-observed function value, $\text{EI}(\mathbf{x}) := \mathbb{E}\left[\max(y^* - f(\mathbf{x}), 0)\right]$, where the expectation is taken over the posterior distribution predicted by the surrogate models, and $y^*$ represents the current best-observed function value, given by $y^* := f(\mathbf{x}_{\min})$ (Frazier, 2018). The EI function

quantifies the expected improvement in the objective value compared to the current best, thereby encouraging the selection of points that are likely to yield better performance.

Equipped with EI, we proceed by optimizing EI over the parameter space to find the most promising point to evaluate, using gradient-based methods such as L-BFGS-B (Zhu et al., 1997). However, motivated by the fact that the parameter space is discrete over parameter counts ($m$) and training steps ($z$), we choose to optimize EI over each unique tuple $(m, z)$. Then, to account for the fact that evaluation for each tuple incurs varying costs $c(m, z)$, we evaluate the point that has the greatest EI per unit cost (EIpu) (Lee et al., 2020). Additional details of our implementation are provided in Appendix C.

### 4.3 Baselines Overview

**Multi-fidelity Bayesian Optimization:**   We employ Hyperband, implemented via SMAC (Lindauer et al., 2022), which leverages a random forest surrogate model and expected improvement as its acquisition function. Hyperband (Li et al., 2018b) efficiently explores multiple parameter configurations by early stopping poorly performing candidates. In our experiments, we fix the model scale at the target model size throughout optimization, which is not cost-aware.

**Random Search:**   We uniformly sample data proportions and evaluate them at the largest model size and maximum training steps. See Appendix D for the detailed algorithm and implementation specifics.

## 5   Results

In this section, we demonstrate the effectiveness of our MFMS-BO framework by benchmarking the simple MFMS-GP method against baselines that do not leverage smaller model scales.

For each of the algorithms evaluated, we run 20 experiments over different seeds, and show the one standard deviation bound with shaded regions. The number of evaluations (the x-axis) was recorded by Wandb (Biewald, 2020) in the experiments we used to train our predictor, and normalized so that 1 evaluation corresponds to the number of flops required to train a 1B model for 100 training steps. As an example, in a "random search" algorithm, where we randomly sample parameters and train models for 20,000 steps wtih those parameters, doing this with five different set of parameters for a 1B model would involve 1,000 evaluation units. In a more complex algorithm, where the number of training steps and model scales are dynamically determined, we report the exact number of evaluation steps that would have been required for the various training runs in that algorithm. The MFMS-GP method requires tuning its hyperparameters. As described in Appendix C, we randomly select a few configurations to tune these parameters, and the cost of evaluating these configurations is accounted for.

Since MFMS-GP relies on GP posterior and potentially noisy EI optimizations to select model scales and training steps, it may take some time before the algorithm samples points at the target scale and fidelity, even after identifying a performant data mixture. If we were to plot only the performance of the best model trained by MFMS-GP at any given evaluation unit, the result could appear deceptively poor simply because the model was smaller or trained for fewer steps, failing to reflect the true quality of the corresponding data mixture. Therefore, to allow us to compare MFMS-GP with other baselines that always train at target model scale, we add an additional curve: **MFMS-GP full-scale**, which shows the performance one would have gotten if one takes the best configuration MFMS-GP has observed, and simply sets the model scale and training steps to the target $m^*$ and $z^*$. The additional curve can be viewed as representing a realistic use of our method: a fixed compute budget is first allocated for exploration, allowing the algorithm to identify the best data mixture. In a subsequent phase, one would take this mixture to train a model to completion at the target scale. The additional computational cost of the final training is of course accounted for in these plots.

In Figures 3 and 4, we see that both of the plots for our MFMS-GP algorithm have a **2.6x to 3.3x** speedup in finding the configuration that achieves the highest accuracy. Observe that MFMS-GP rapidly explores different data mixtures and identifies promising ones earlier than Hyperband or random search. This is expected, since MFMS-GP can utilize inexpensive evaluations by training on smaller models and for fewer steps. These strong results — achieved with a simple GP-based method

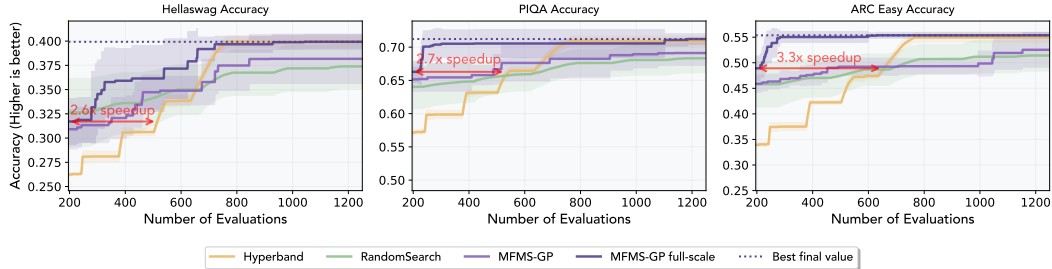

Figure 3: On maximizing accuracy in the downstream tasks, our multi-scale multi-fidelity approach achieves more than 2.6x speedup and finds the best configuration the fastest. Shaded area indicates standard deviation across random seeds.

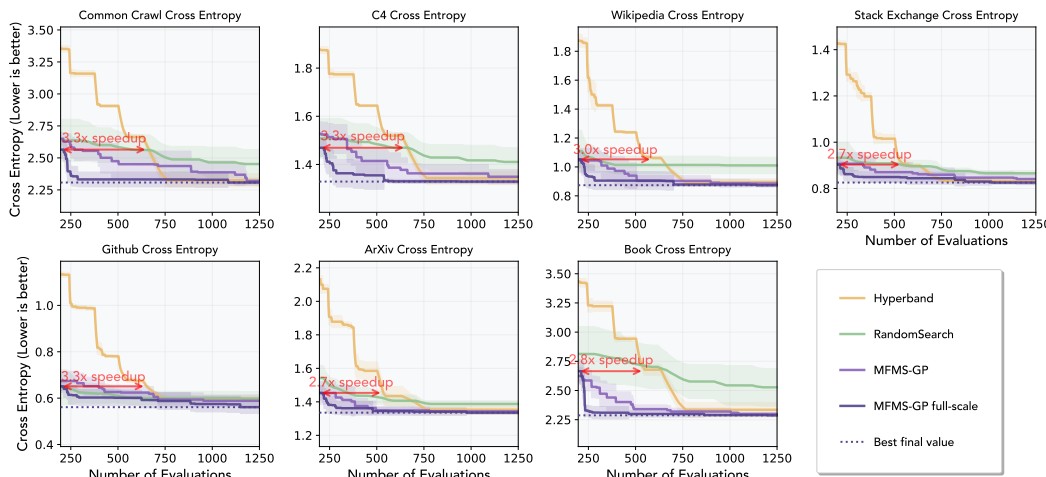

Figure 4: On minimizing the validation cross-entropy losses, our multi-scale multi-fidelity approach achieves more than 2.6x speedup and finds the best configuration the fastest. Shaded area indicates standard deviation across random seeds.

— underscore the potential of the MFMS framework in efficiently gathering low-cost yet valuable information to guide large-scale language model training. We expect that thoughtful algorithmic improvements, such as better kernel design or acquisition functions, will lead to even more powerful methods. We include the results using several standard kernels in Appendix E, and leave more extensive explorations of this new setting to future work.

## 6 Related Work

**Data Mixtures.** Several approaches aim to move beyond heuristic methods for data mixture by leveraging algorithmic techniques. Albalak et al. (2023b) propose an online data mixing strategy using a non-stochastic bandit algorithm to dynamically adjust data proportions during training, maximizing perplexity. DoReMi (Xie et al., 2023) focuses on identifying and emphasizing the "hardest" datasets for a base model through distributionally robust language modeling to improve training efficiency. Ge et al. (2025b) model a joint scaling behavior of domain proportions and training steps; we push this further through modeling the model scale. Goyal et al. (2024b) delve into the quality-quantity tradeoff in data, exploring how data filtering and repetition affect model performance and introducing scaling laws that account for data utility decay. These works highlight the increasing interest in principled and adaptive methods for data mixture optimization, yet often focus on fixed model scales. In contrast, our multi-fidelity multi-scale approach considers the practical scenario where practitioners have a fixed budget to experiment with data mixture, and can exploit cheap information gathered from smaller models and early training steps.

**Scaling Laws.** Scaling laws provide crucial insights into the relationship between model size, training compute, and performance in large language models (Kaplan et al., 2020; Li et al., 2025; Pearce and Song, 2024; Zhao et al., 2025; Mikami et al., 2021; Aghajanyan et al., 2023). Hoffmann et al. (2022b) established foundational scaling laws demonstrating predictable performance improvements with increased compute, model parameters, and training data. (Muennighoff et al., 2023) investigates the impact of data repetition in data-constrained scenarios, showing diminishing returns beyond a certain repetition threshold. Ruan et al. (2024) propose observational scaling laws based on "principal capabilities" to explain and predict language model performance across diverse models and benchmarks, as has been done in other BO settings. These scaling law studies motivate our framework by providing an empirical basis for extrapolating performance variations of different data mixtures across model scales.

**Bayesian Optimization.** Data mixture optimization, like hyperparameter tuning, benefits from efficient search strategies. Approaches range from full configuration selection with methods like BO to hyperband, which employs early termination of unpromising runs. Early BO methods (Hutter et al., 2011) used GPs to model the relationship between hyperparameters and model performance. Subsequent works explored random forests (Lindauer et al., 2022) and Parzen estimators (Bergstra et al., 2011) as surrogate models. Early stopping techniques like Hyperband (Li et al., 2018b) focus on efficiently evaluating multiple parameter configurations by progressively eliminating poorly performing candidates, and exploring many combinations with fewer resources. More recent methods like BOHB (Falkner et al., 2018) combine both the BO exploration of Parzen estimators with the multi-fidelity benefits of Hyperband. Our work, MFMS-BO, is the *first* to explore a multi-fidelity multi-scale approach for data mixture optimization.

## 7 Conclusion and Future Work

This work introduces a principled framework, multi-fidelity multi-scale Bayesian optimization, for optimizing data mixture compositions in large language model training, a critical challenge in modern AI system development. Our framework unifies recent advances in predicting optimal data mixtures across scales with classical multi-fidelity BO techniques. Based on this unified framework, we implemented the GP using the RBF kernels and expected-improvement-per-unit acquisition function to balance the information gain and the cost of exploring new points in the functional landscape. We find that the method achieves optimal downstream task performance **2.6 times faster** than traditional multi-fidelity approaches by strategically exploring the joint space of data mixtures and model scales.

In addition, we empirically demonstrate two key insights that inform future efficient optimization of data mixtures. First, our analysis reveals that training runs on smaller models (below 500M parameters) provide valuable predictive signals for optimizing larger architectures (1B parameters). Second, we establish that partial training runs can effectively inform full-scale training decisions. Specifically, our results show that a combination of full and partial training runs (e.g., 5 complete and 10 half-length runs) yields better predictive utility than an equal-compute allocation of full training runs alone (e.g., 10 complete runs).

Several promising directions emerge for future research. First, extending our framework to more settings, such as language model fine-tuning, data filtering, and diverse collections of datasets with more categories, would help validate its generalizability across different data mixing scenarios. In particular, the low-cost fine-tuning regime provides an opportunity to evaluate the framework without the predictor. From a methodological perspective, incorporating domain knowledge about the positive correlation between model performance and both parameter count and training duration could enhance GP kernel design. Exploring alternative acquisition functions, such as knowledge gradient (Poloczek et al., 2016; Wu et al., 2019), could further improve efficiency in navigating the optimization landscape. The fundamental differences between model scale and training steps as fidelity dimensions also require further investigation to refine their treatment within the framework. Other realistic considerations could enhance the practical benefits of the framework, such as developing algorithms supporting batch evaluations and asynchronous updates for efficient parallel exploration. Addressing these challenges will further strengthen the framework and its applicability in large-scale language model training.

## Acknowledgements

The authors thank Empire AI (Bloom et al., 2025) for generously providing extensive computational resources, without which this study would not be possible. This work was supported by the Digital Future Initiative at Columbia Business School.

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

# Appendix

## A    Pretraining Runs Details

We use the OLMo 2 OLMo et al. (2024) package for training our language models. The model configurations are

| Group | d_model | n_heads | n_layers | Runs |
|-------|---------|---------|----------|------|
| 20M   | 256     | 8       | 8        | 115  |
| 60M   | 512     | 8       | 8        | 71   |
| 150M  | 768     | 12      | 12       | 53   |
| 300M  | 1024    | 16      | 16       | 74   |
| 500M  | 1280    | 16      | 16       | 39   |
| 700M  | 1536    | 16      | 16       | 52   |
| 1B    | 2048    | 16      | 16       | 68   |

Table 4: Model Architecture Details by Group with Number of Runs

To decide the training mixtures of the experiments, for each experiment, we randomly sample from the probability simplex using the Dirichlet distribution of order $n$ (number of training datasets), where we use the parameters $\alpha_i = 1, \forall i \in [n]$. Other training configurations (learning rate, momentum, etc.) are directly taken from OLMo's configuration files (e.g. 700M) We study the compute optimal regime (Hoffmann et al., 2022a): for each 1B model run, we used 20B tokens in total for training. In the interest of collecting more runs, all other model scales are trained on 10B tokens.

## B    Predictor Training

We train predictors using multilayer perceptrons (MLPs) consisting of three hidden layers with 64 hidden units each, ReLU activations, and dropout with a rate of 0.1, totaling approximately 5,000 parameters. The predictors accept three types of inputs: (1) the model size (number of parameters), (2) the number of training steps, and (3) the proportions of each of the five dataset categories used during pretraining, namely (*Wikipedia*, *StackExchange*, *Github*, *ArXiv*, and *Book*).

For each pretrained language model, the predictor outputs predictions for multiple metrics: the training loss, validation losses across seven categories (*Wikipedia*, *StackExchange*, *Github*, *ArXiv*, *Book*, and held-out datasets *CommonCrawl* and *C4*), as well as downstream task accuracies on three evaluation tasks: *HellaSwag*, *PIQA*, and *Arc Easy* (Zellers et al., 2019; Bisk et al., 2020; Clark et al., 2018). Thus, each language model corresponds to a single row in our predictor's dataset, comprising 9 inputs as described above and outputs spanning these 11 metrics.

We train the predictors to maximize the coefficient of determination ($R^2$), $R^2 = 1 - \frac{\sum_i (y_i - \hat{y}_i)^2}{\sum_i (y_i - \bar{y})^2}$, where $y_i$ is the true metric value for data point $i$, $\hat{y}_i$ is the predictor's estimate, and $\bar{y}$ is the mean of true values. Training of our predictor is conducted for 20 epochs using a batch size of 64, an Adam optimizer with a learning rate of 0.001 and weight decay of 0.01, and data normalization (standard scaling) applied to both inputs and outputs.

## C    Bayesian Optimization Details

For MFMS-GP, the cost of evaluating a run at a particular model scale is taken from the number of FLOPS the corresponding model scale costs during the pretraining runs. The costs are scaled appropriately such that a unit of cost corresponds FLOPS required to train 1B model for 1 training step.

Additionally, since it is prohibitively expensive to optimize EI for each of 19700 training steps, for multi-fidelity multi-scale GP, we limit the space of training steps to be $\mathcal{Z} = \{6000, 12000, 19700\}$.

To initiate the hyperparameters of MFMS-GP, we randomly select 20 configurations up to training step $z = 9$ to fit the kernel and mean functions' parameters. The GP hyperparameters are trained using the Adam optimizer (Kingma and Ba, 2014) with a 0.1 learning rate for 50 iterations.

As mentioned in Section 4.2, we optimize EI within $(m, z)$ tuple. For this optimization, we initiate 5 random probability weights drawn from the Dirichlet distribution and perform a gradient search for greater EI over the probability simplex.

Occasionally, the GP would be too certain of its posterior prediction such that the optimized EIs are all small in magnitude. Therefore, when the optimal EI is below a certain threshold, we lower the length scales of the RBF kernels to encourage more exploration. The threshold is set to be $1e^{-4}$, and the length scales would be lowered to $95\%$ of their original values.

As a measure to encourage selecting higher cost evaluations later in the optimization cycle, instead of using $\text{EI}_{\text{pu}}(\lambda) = \frac{\text{EI}(\lambda)}{c(m,z)}$, we introduce an additional parameter $\alpha$ that controls the importance of cost, and pick the configuration that maximizes $\frac{\text{EI}(\lambda)}{c(m,z)^\alpha}$. Initially $\alpha = 1$, and it decays by $1\%$ for every step of the Bayesian optimization.

## D Baselines

### D.1 Baselines: Multi-Fidelity Bayes Opt

---
**Algorithm 2** Hyperband with Random Forest, EI

---
1: **Input:** Probability space $\Delta^n$, training-step space $\mathcal{Z}$, target model scale $m^*$, and cost function $c(m^*, \cdot)$
2: Initialize random forest surrogate model RF
3: Set initial design with Random Sampling
4: Initialize history $\mathcal{H} = \emptyset$
5: **for** each Hyperband iteration **do**
6:     Split the total computation budget into $s$ brackets
7:     **for** each bracket $s_i$ **do**
8:         Generate initial configurations $\boldsymbol{w}_1, \ldots, \boldsymbol{w}_n$ at lowest fidelity $z = 1$
9:         **for** each fidelity $z$ from 1 to $z^*$ **do**
10:             Evaluate configurations $\boldsymbol{w}_i$ at fidelity $z$
11:             Store results in $\mathcal{H}$
12:             Fit RF on $\mathcal{H}$
13:             Select next $\lambda_{\text{next}}$ using Expected Improvement
14:             Update $\mathcal{H}$ with new evaluations
15:         **end for**
16:     **end for**
17: **end for**
18: Return best $\boldsymbol{w}^* = \lambda^*[0]$ from configuration $\lambda^* = \arg\max_{\lambda \in \mathcal{H}} \mu(\lambda)$

---

As a baseline for multi-fidelity Bayesian optimization, we use Hyperband implemented by SMAC: Sequential Model-Based Optimization for General Algorithm Configuration (Lindauer et al., 2022). This multi-fidelity Bayesian optimization uses a random forest as a surrogate model, expected improvement as the acquisition function, and uses Hyperband (Li et al., 2018b), which is an early stopping technique that focuses on efficiently evaluating multiple parameter configurations by progressively eliminating poorly performing candidates, and exploring many combinations with fewer resources. Since the multi-fidelity framework does not offer a straightforward way to incorporate the additional dimension of model scale, throughout the optimization, we fix the number of the model's parameters to the target model scale $m^*$.

### D.2 Baselines: Random Search

For a baseline that does not consider utilizing the fidelity dimension, we consider random search. Random search selects data proportions that are uniformly drawn from our data proportion space. We then run it against the largest model size and training steps.

# E    Kernel Comparisons

The kernel function $k(x, x')$ in a GP defines the covariance between different input points. In the main results, we use a RBF kernel $k(\boldsymbol{w}, \boldsymbol{w}') = \exp(-\frac{d(\boldsymbol{w}, \boldsymbol{w}')}{2\sigma^2})$ for the data proportions $\boldsymbol{w}$, where $d(\boldsymbol{w}, \boldsymbol{w}') = \|\boldsymbol{w} - \boldsymbol{w}'\|^2$ is the squared $L^2$ distance between the two probabilities. In this appendix, we experiment with distance metrics that may be more suitable for probabilities.

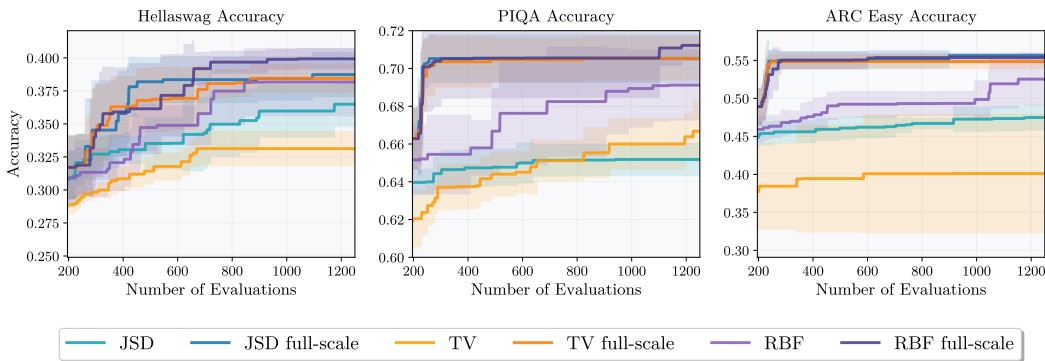

Figure 5: Comparing different Gaussian process kernels on maximizing accuracy in the downstream tasks.

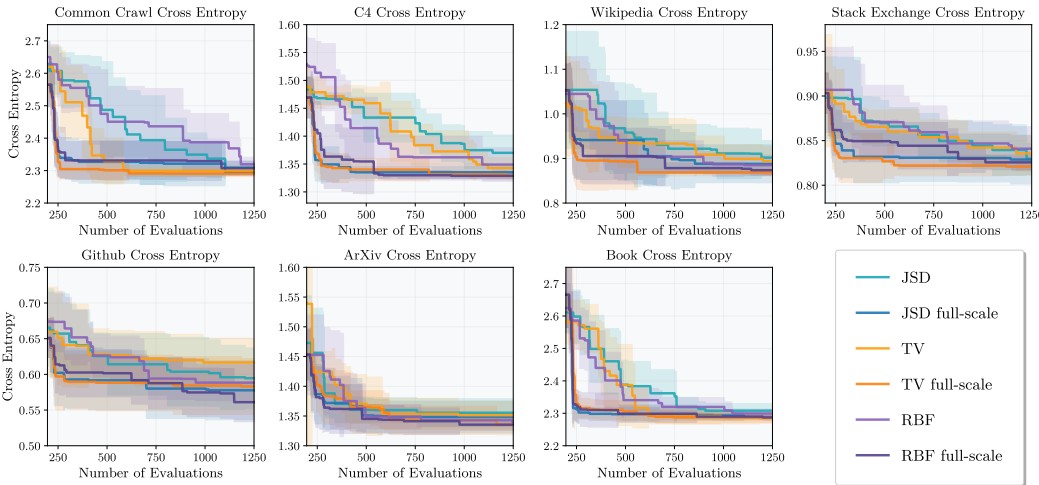

Figure 6: Comparing different Gaussian process kernels on minimizing the validation cross-entropy losses

Specifically, we consider the Total Variance (TV) distance and the Jensen–Shannon divergence (JSD), a symmetric Kullback–Leibler divergence, for the data proportions:

$$\text{TV}: \qquad d(\boldsymbol{w}, \boldsymbol{w}') = \|\boldsymbol{w} - \boldsymbol{w}'\|_1$$

$$\text{JSD}: \qquad d(\boldsymbol{w}, \boldsymbol{w}') = \frac{1}{2}\text{KL}\left(\boldsymbol{w} \parallel \bar{\boldsymbol{w}}\right) + \frac{1}{2}\text{KL}\left(\boldsymbol{w}' \parallel \bar{\boldsymbol{w}}\right)$$

where $\text{KL}(\cdot)$ denotes the Kullback–Leibler divergence, and $\bar{\boldsymbol{w}} = \frac{\boldsymbol{w} + \boldsymbol{w}'}{2}$.

We found that on optimizing accuracy, the simple RBF kernels generally perform better. However, occasionally, JSD (on HellaSwag of Figure. 5) or TV (on Common Crawl, Wikipedia, and Stack Exchange of Figure. 6) yield better results.

