# OpenReview forum: "Data Mixture Optimization: A Multi-fidelity Multi-scale Bayesian Framework"
_NeurIPS.cc/2025/Conference — NeurIPS 2025 poster_

### Official Review · Reviewer_FwDJ · 2025-06-30

**Clarity:** 3
**Significance:** 3
**Originality:** 4
**Rating:** 5
**Confidence:** 3

**Summary:**

The paper proposes an approach to data curation for training LLMs based on
multi-fidelity multi-scale BO. The authors describe their approach and evaluate
it empirically, demonstrating speedups in training.

**Questions:**

See above.

**Ethical Concerns:**

["NO or VERY MINOR ethics concerns only"]

**Final Justification:**

This paper presents a nice idea that works well in practice and should be accepted in my opinion.

**Limitations:**

Yes.

**Quality:**

3

**Strengths And Weaknesses:**

The paper is well written and proposes an interesting idea that addresses one of
the main challenges in training LLMs. To the best of my knowledge, the idea is
novel and, while relatively simple, seems to work well in practice. The proposed
method does not necessarily result in better predictive performance, but enables
the training process to achieve a particular level of performance faster, which
is especially important in the context of large models such as LLMs.

It was unclear to me why the authors used MLPs as predictor models for
estimating the performance of particular combinations of model size, dataset
proportions etc. MLPs are not commonly used in BO, and later in the paper the
authors use the more common GPs as surrogate models. Did the authors evaluate
different approaches and settle on MLPs because of better performance? Please
clarify.

The empirical evaluation relies exclusively on surrogate models instead of
actual performance data as far as I can see. While the overall R^2 seems good,
the results for individual cases vary quite a bit (Table 3), especially for
smaller fractions of the overall data. The authors say that results were
averaged over 3 random seeds, but do not give confidence intervals. A (small)
additional experiment on real data would help to convince the reader that the
presented results hold true in practice.

---

> ### Author Rebuttal · Authors · 2025-07-31
>
> # General Response
>
> We sincerely thank the reviewers for their thoughtful feedback and careful evaluation of our work. We are grateful that our paper was well received with positive scores (5, 5, 4, 3). Particularly, we are encouraged that the reviewers appreciate the novelty of our approach (6aWf, H7gH, bchD, FwDJ), the importance of the problem tackled in this work (6aWf, H7gH, bchD, FwDJ), and the performance of MFMS-GP (H7gH, bchD, FwDJ).
>
>
> Reviewer bchD raised valid concerns regarding our use of the simulator and existing Bayesian Optimization (BO) methods. In response, we would like to take this opportunity to clarify and highlight what we view as the main contributions of our work:
> - (a) providing the simulator as a testbed for methodological development (echoing Reviewer 6aWf’s comment)
> - (b) proposing a sequential decision making framework that naturally describes data mixture optimization.
>
> (a) A clear bottleneck of developing methods for data mixture optimization is the high cost of training new LLMs each time a method proposes a different mixture. To make progress, the scientific community needs a high-fidelity but cheap setting in order to develop effective methodologies. Therefore, we decided to build a novel high-fidelity simulator suitable for testing and benchmarking principled algorithms for data mixing. We believe this simulator will be a useful resource for the community, supporting future advancements in this area. We would also like to note that the use of a simulator is a common approach in the study of hyper-parameter optimization where evaluation is expensive [1, 2].
>
> (b) Our motivation for proposing the sequential decision framework stems from the current status-quo in the literature: the existing approaches to data mixing optimization rely on (i) ad-hoc data curation approaches and (ii) standard scaling laws that perform deterministic extrapolation. While using BO for the sequential decision framework may seem natural, its actual effectiveness has not been established in prior work, leaving practitioners either unaware of its potential or hesitant to adopt it. We are among the first to rigorously demonstrate the effectiveness of BO for data-mixing optimization, and this verification is far from trivial—it required training 472 models up to 1B parameters. We hope these promising results will spur further exploration of BO for data mixing and other pretraining optimizations, potentially leading to significant savings in both cost and compute.
>
>
> We hope to address your questions and concerns satisfactorily this round, and are truly thankful for your feedback, which has helped improve the paper in various ways.
>
> [1] Zela et al., Surrogate NAS Benchmarks: Going Beyond the Limited Search Spaces of Tabular NAS Benchmarks, International Conference on Learning Representations, 2022
>
> [2] Eggensperger et al., Efficient benchmarking of hyperparameter optimizers via surrogates, Proceedings of the Twentynineth National Conference on Artificial Intelligence, 2015
>
> # Point-by-Point Responses to Reviewer FwDJ
> We are thankful for Reviewer FwDJ’s positive perspective on the paper, and are grateful that the reviewer appreciates the novelty of the idea and the performance of our method.
>
> >**It was unclear to me why the authors used MLPs as predictor models for estimating the performance of particular combinations of model size, dataset proportions etc. MLPs are not commonly used in BO**
>
> We apologize for the confusion caused by our exposition. The MLP in fact serves only as a surrogate benchmark on which we validate our BO method (MFMS-GP), and is not used in the BO. Since training a new language model for every hyperpameter sets that BO methods recommend would be prohibitively expensive, we chose to train a MLP model. We then benchmarked MFMS-GP against baselines on the output generated by the MLP.
>
> >**While the overall R^2 seems good, the results for individual cases vary quite a bit (Table 3)... A (small) additional experiment on real data …**
>
> We would like to highlight that the results in Table 3 artificially limits the training dataset available to the MLPs (to demonstrate the predictive value of smaller models’ results), and do not reflect the quality of the MLP that was used as the surrogate benchmark. For the main results of the paper, we used the MLP trained on 422 real language model training runs, and validated the MLP on 50 real validation runs. The MLP achieves a R^2 greater than 0.95 across all metrics validated on real data.

---

> > ### Comment · Reviewer_FwDJ · 2025-07-31
> >
> > Thank you for the clarifications!

---

### Official Review · Reviewer_bchD · 2025-07-03

**Clarity:** 3
**Significance:** 3
**Originality:** 2
**Rating:** 4
**Confidence:** 4

**Summary:**

This paper proposes a multi-fidelity multi-scale Bayesian Optimization framework for optimizing data mixture compositions in LLM training. Instead of relying on intuition or costly trial-and-error approaches, the method treats data curation as a sequential decision-making problem that adaptively selects data mixtures, model scales, and training steps to balance cost and information gain. Using a simulator trained on 472 language model runs, the authors show their GP-based approach can leverage cheap evaluations from smaller models to guide decisions for larger ones, achieving 2.6x to 3.3x speedups compared to baseline methods while finding optimal data mixtures for models from 20M to 1B parameters.

**Questions:**

See weaknesses.

**Ethical Concerns:**

["NO or VERY MINOR ethics concerns only"]

**Final Justification:**

The authors have addressed most of my concerns, especially the one about the pre-trained models.

**Limitations:**

No. The authors claim they have included Limitations in the last section "Conclusions and Future Work", but to me it's not really a discussion of the current work's limitation.

**Quality:**

3

**Strengths And Weaknesses:**

**Strength**
1. **Novel problem formulation:** First to treat data mixture optimization as a multi-fidelity multi-scale BO problem, providing a principled alternative to heuristic approaches.
2. **Practical applicability: This paper addresses real-world computational constraints faced by practitioners training LLMs with limited budgets**.
3. **Significant performance gains:** It achieves 2.6x to 3.3x speedups compared to baseline methods across multiple evaluation metrics, although it was achieved on a simulator benchmark.

**Weaknesses**
1. **Limited scale and scope:** The paper only tested on models up to 1B parameters and five data categories from SlimPajama, raising questions about generalizability to larger, production-scale models and diverse datasets.
2. **Limited practical validation:** I know it might be a bit too much to ask the authors to perform BO on the full LLM training, but training an MLP on 472 records and using it as the simulator for validation has serious limitation. There is no demonstration on real-world deployment scenarios or comparison with industry-standard data curation practices, which can severely qualify the significance of the proposed claims.
3. **Limited technical novelty:** It is kind of natural to treat the data mixing ratio as hyperparameters of the training framework and model this problem as BO. The current approach largely combines existing BO (standard GP), without introducing significant algorithmic innovations or theoretical contributions.

---

> ### Author Rebuttal · Authors · 2025-07-31
>
> # General Response
>
> We sincerely thank the reviewers for their thoughtful feedback and careful evaluation of our work. We are grateful that our paper was well received with positive scores (5, 5, 4, 3). Particularly, we are encouraged that the reviewers appreciate the novelty of our approach (6aWf, H7gH, bchD, FwDJ), the importance of the problem tackled in this work (6aWf, H7gH, bchD, FwDJ), and the performance of MFMS-GP (H7gH, bchD, FwDJ).
>
>
> Reviewer bchD raised valid concerns regarding our use of the simulator and existing Bayesian Optimization (BO) methods. In response, we would like to take this opportunity to clarify and highlight what we view as the main contributions of our work:
> - (a) providing the simulator as a testbed for methodological development (echoing Reviewer 6aWf’s comment)
> - (b) proposing a sequential decision making framework that naturally describes data mixture optimization.
>
> (a) A clear bottleneck of developing methods for data mixture optimization is the high cost of training new LLMs each time a method proposes a different mixture. To make progress, the scientific community needs a high-fidelity but cheap setting in order to develop effective methodologies. Therefore, we decided to build a novel high-fidelity simulator suitable for testing and benchmarking principled algorithms for data mixing. We believe this simulator will be a useful resource for the community, supporting future advancements in this area. We would also like to note that the use of a simulator is a common approach in the study of hyper-parameter optimization where evaluation is expensive [1, 2].
>
> (b) Our motivation for proposing the sequential decision framework stems from the current status-quo in the literature: the existing approaches to data mixing optimization rely on (i) ad-hoc data curation approaches and (ii) standard scaling laws that perform deterministic extrapolation. While using BO for the sequential decision framework may seem natural, its actual effectiveness has not been established in prior work, leaving practitioners either unaware of its potential or hesitant to adopt it. We are among the first to rigorously demonstrate the effectiveness of BO for data-mixing optimization, and this verification is far from trivial—it required training 472 models up to 1B parameters. We hope these promising results will spur further exploration of BO for data mixing and other pretraining optimizations, potentially leading to significant savings in both cost and compute.
>
>
> We hope to address your questions and concerns satisfactorily this round, and are truly thankful for your feedback, which has helped improve the paper in various ways.
>
> [1] Zela et al., Surrogate NAS Benchmarks: Going Beyond the Limited Search Spaces of Tabular NAS Benchmarks, International Conference on Learning Representations, 2022
>
> [2] Eggensperger et al., Efficient benchmarking of hyperparameter optimizers via surrogates, Proceedings of the Twentynineth National Conference on Artificial Intelligence, 2015
>
> # Point-by-Point Responses to Reviewer bchD
>
> We thank the reviewer for finding our approach novel and practical. The limitations raised by the reviewer are indeed compromises we made under the constraint of academic budget. We hope that our responses clarify the rationale behind these choices and demonstrate how they do not undermine the key scientific contributions of the work.
>
> >**...raising questions about generalizability to larger, production-scale models and diverse datasets**
>
> To build the simulator that supports the study, we had to train a total of 472 language models, including 68 models at the 1B scale. Unfortunately, training a comparable number of significantly larger models was beyond our available computational budget.
> Similarly, expanding the number of domains would have required substantially more training runs at each model scale to cover the larger space of data mixtures. We chose to focus on the RedPajama dataset with five training categories as a result. That said, we expect our approach to scale effectively to ~20 domains, as commonly observed for a Bayesian optimization method.
> >**might be a bit too much to … perform BO on the full LLM training ...simulator for validation has serious limitation…**
>
> We agree with the reviewer that performing BO without a simulator would be prohibitively expensive, and we also recognize that using a simulator introduces limitations. However, we would like to highlight that use of simulators, sometimes referred to as surrogate benchmarking [1, 2], is a well-established strategy commonly used to study hyperparameter-tuning of expensive-to-train models.
>
> [1] Zela et al., Surrogate NAS Benchmarks: Going Beyond the Limited Search Spaces of Tabular NAS Benchmarks, International Conference on Learning Representations, 2022
>
> [2] Eggensperger et al., Efficient benchmarking of hyperparameter optimizers via surrogates, Proceedings of the Twentynineth National Conference on Artificial Intelligence, 2015
>
>
> >**It is kind of natural to treat the data mixing ratio as hyperparameters… current approach largely combines existing BO…**
>
> We in fact view the naturalness of the BO approach as a strength. While natural, the approach has been overlooked in the current literature on data mixture optimization. By formulating and empirically validating this framework without developing a tailored BO method, we demonstrate its potential and aim to encourage its broader adoption in future research. We believe this is a meaningful contribution in its own right.
>
> >**No. The authors claim they have included Limitations in the last section "Conclusions and Future Work", but to me it's not really a discussion of the current work's limitation.**
>
> We thank the reviewer for the feedback that has inspired valuable discussions. We'll make sure to include the discussions and thoroughly address the aforementioned limitations with the extra page afforded to us in the camera ready version.

---

> > ### Comment · Reviewer_bchD · 2025-08-01
> > **Thank you for your response**
> >
> > Thank you for your response.
> >
> > The rebuttal has addressed my concerns and clarified my misunderstanding that the >400 models were pre-trained by the authors rather than being part of the released records in OLMO2. I just have a quick follow-up question regarding the release of the trained models.
> >
> > I have adjusted my score accordingly.

---

> ### Author Response · Authors · 2025-08-03
> **Thank you!**
>
> We thank the reviewer for their consideration, and we’re happy that the response addressed your concerns. We plan to release the trained models, as well as all the accuracies, losses and other metadata obtained during training.

---

### Official Review · Reviewer_H7gH · 2025-07-03

**Clarity:** 3
**Significance:** 3
**Originality:** 3
**Rating:** 4
**Confidence:** 4

**Summary:**

This paper tackle with a critical and challenging problem of data mixture optimization for LLM pretraining. The authors propose a novel MFMS-BO method to apply bayesian optimization to predict the optimal data mixture composition from multiple data sets from previous training runs with various training steps and model scales. The empirical results on the SlimPajama datasets demonstrate the proposed MFMS BO method can estimate the optimal data mixture for larger-scale models from smaller-scale experiments, which achieves up to 3 times speedup above the random sampling baseline.

**Questions:**

1. Lack of comparisons to some critical baseline approaches. could you provide the comparison results to Regmix [1] and data mixing law [2], where only various model scales are considered?

2. The benchmark is only limited to SlimPajama with 5 training domains. The scalability and effectiveness of the MFMS-BO method on larger number of domains has not been explored, e.g. The pile (22 domains) or hundreds of domains clustered from document embeddings.

3. Can you provide a discussion on the computation complexity of the proposed method, and compare with traditional methods like DML and RegMix?

4. How many simulation runs are required to obtain a relatively accurate estimation on the optimal data mixture?

5. The current out-of-domain extrapolation results on the MFMS-BO method is limited, since CC and C4 are known as general web-crawled dataset containing diverse topics. Provide the results on more domain-specific datasets (e.g. freelaw, dm-mathematics in The Pile dataset) could help to improve the understanding on its OOD performance and strengthen the paper.


[1] RegMix: Data Mixture as Regression for Language Model Pre-training.

[2] Data Mixing Laws: Optimizing Data Mixtures by Predicting Language Modeling Performance.

**Ethical Concerns:**

["NO or VERY MINOR ethics concerns only"]

**Final Justification:**

This paper tackles with data mixture optimization for LLM pretraining and proposes a novel MFMS-BO method to apply bayesian optimization to predict the optimal data mixture composition from multiple data sets from previous training runs with various training steps and model scales. The empirical results on the SlimPajama datasets demonstrate the proposed MFMS-BO method can estimate the optimal data mixture for larger-scale models from smaller-scale experiments, which achieves up to 3 times speedup above the random sampling baseline.

During rebuttal, the authors provide additional results comparing to other data mixing baselines, e.g. RegMIX, which demonstrate slight performance gain over RegMix. I believe that the proposed MFMS-BO method can be an efficient and effective alternative to the traditional scaling law studies.

**Limitations:**

1. Lack of comparisons to some critical baseline approaches. For example, the Regmix [1] and data mixing law [2], where the optimal data mixture is predicted on experiments across various model scales, should be considered as baselines.

2. The benchmark is only limited to SlimPajama with 5 training domains. The scalability and effectiveness of the MFMS-BO method on larger number of domains has not been explored? e.g. The pile (22 domains) or hundreds of domains clustered from document embeddings.

3. The out-of-domain extrapolation results on the MFMS-BO method is limited, since CC and C4 are known as general web-crawled dataset containing diverse topics. Provide the results on more domain-specific datasets (e.g. freelaw, dm-mathematics in The Pile dataset) could help to improve the understanding on its OOD performance and strengthen the paper.

**Paper Formatting Concerns:**

no major concerns

**Quality:**

3

**Strengths And Weaknesses:**

### Strength
1. The paper tackles a critical and challenging problem of data mixture optimization in LLM pretraining.
2. This paper introduce a critical consideration of data mixture optimization, the *fidelity* (length) of the small-scale training runs, which has been ignored by previous law-fitting based methods, such as RegMix [1] or Data Mixing Law [2].
3. A novel probabilistic extrapolation framework framework is proposed based on bayesian optimization, which explicitly consider the uncertainty and fidelity of small-scale experiments and derive a more accurate estimation of the optimal data mixture in larger-scale pretraining experiments. The empirical results demonstrate a large speedup compared to random sampling baselines.

### Weakness
1. Lack of comparisons to some critical baseline approaches. For example, the Regmix [1] and data mixing law [2], where the optimal data mixture is predicted on experiments across various model scales, should be considered as baselines.

2. The benchmark is only limited to SlimPajama with 5 training domains. The scalability and effectiveness of the MFMS-BO method on larger number of domains has not been explored? e.g. The pile (22 domains) or hundreds of domains clustered from document embeddings.

3. The out-of-domain extrapolation results on the MFMS-BO method is limited, since CC and C4 are known as general web-crawled dataset containing diverse topics. Provide the results on more domain-specific datasets (e.g. freelaw, dm-mathematics in The Pile dataset) could help to improve the understanding on its OOD performance and strengthen the paper.

[1] RegMix: Data Mixture as Regression for Language Model Pre-training.

[2] Data Mixing Laws: Optimizing Data Mixtures by Predicting Language Modeling Performance.

---

> ### Author Rebuttal · Authors · 2025-07-31
>
> # General Response
>
> We sincerely thank the reviewers for their thoughtful feedback and careful evaluation of our work. We are grateful that our paper was well received with positive scores (5, 5, 4, 3). Particularly, we are encouraged that the reviewers appreciate the novelty of our approach (6aWf, H7gH, bchD, FwDJ), the importance of the problem tackled in this work (6aWf, H7gH, bchD, FwDJ), and the performance of MFMS-GP (H7gH, bchD, FwDJ).
>
>
> Reviewer bchD raised valid concerns regarding our use of the simulator and existing Bayesian Optimization (BO) methods. In response, we would like to take this opportunity to clarify and highlight what we view as the main contributions of our work:
> - (a) providing the simulator as a testbed for methodological development (echoing Reviewer 6aWf’s comment)
> - (b) proposing a sequential decision making framework that naturally describes data mixture optimization.
>
> (a) A clear bottleneck of developing methods for data mixture optimization is the high cost of training new LLMs each time a method proposes a different mixture. To make progress, the scientific community needs a high-fidelity but cheap setting in order to develop effective methodologies. Therefore, we decided to build a novel high-fidelity simulator suitable for testing and benchmarking principled algorithms for data mixing. We believe this simulator will be a useful resource for the community, supporting future advancements in this area. We would also like to note that the use of a simulator is a common approach in the study of hyper-parameter optimization where evaluation is expensive [1, 2].
>
> (b) Our motivation for proposing the sequential decision framework stems from the current status-quo in the literature: the existing approaches to data mixing optimization rely on (i) ad-hoc data curation approaches and (ii) standard scaling laws that perform deterministic extrapolation. While using BO for the sequential decision framework may seem natural, its actual effectiveness has not been established in prior work, leaving practitioners either unaware of its potential or hesitant to adopt it. We are among the first to rigorously demonstrate the effectiveness of BO for data-mixing optimization, and this verification is far from trivial—it required training 472 models up to 1B parameters. We hope these promising results will spur further exploration of BO for data mixing and other pretraining optimizations, potentially leading to significant savings in both cost and compute.
>
>
> We hope to address your questions and concerns satisfactorily this round, and are truly thankful for your feedback, which has helped improve the paper in various ways.
>
> [1] Zela et al., Surrogate NAS Benchmarks: Going Beyond the Limited Search Spaces of Tabular NAS Benchmarks, International Conference on Learning Representations, 2022
>
> [2] Eggensperger et al., Efficient benchmarking of hyperparameter optimizers via surrogates, Proceedings of the Twentynineth National Conference on Artificial Intelligence, 2015
>
>
> # Point-by-Point Responses to Reviewer H7gH
>
> We thank the reviewer for your thoughtful comments and for appreciating the novelty of our approach and the significance of the problem.
>
> >**How many simulation runs are required to obtain a relatively accurate estimation on the optimal data mixture?**
>
> We confirmed by grid search that MFMS-GP indeed converged near the optimal data mixture as shown in Figures 3 and 4. Performances on most metrics converged within 500 numbers of evaluations, and all converged near the optimum by 800 evaluations.
>
> >**... could you provide the comparison results to Regmix [1] and data mixing law [2]...**
>
> We would like to highlight that the sequential decision framework under which our method operates affords more flexibility than the explore-then-commit paradigm implicitly adopted by the two referenced approaches. Concretely, it’s evident that in the two mentioned approaches, one needs to arbitrarily determine the scale and the number of the models to be trained to fit a function, and then train one model at the target scale in the end, without clear guidance on which mixture to try next should one have access to more compute. In contrast, our framework naturally considers the selection of samples sequentially from start to end. Therefore, it is more suitable to view the two approaches as possible acquisition functions that one can employ under the framework proposed in this work. That said, we provide direct comparison with the two methods below.
>
> Comparison with Regmix:
>
> |                | Hyperband | RegMix (LightGBM, 512 models at 20M scale) | MFMS-GP full-scale |
> | -------------- | ---- | ------------------ | ------------------ |
> | Common Crawl   | 2.66 | 2.40               | **2.31**           |
> | C4             | 1.52 | 1.35               | 1.33               |
> | Wikipedia      | 1.06 | 0.97               | **0.88**               |
> | Stack Exchange | 1.90 | 0.83               | 0.84               |
> | Github         | 0.67 | 0.56               | 0.59               |
> | ArXiv          | 1.45 | 1.37               | 1.35               |
> | Book           | 2.67 | 2.41               | **2.29**           |
>
> Bolded values indicate the best-performing method, with alternatives falling outside one standard deviation from 15 runs.
> Note that the Regmix results on the three accuracy metrics are considerably worse than Hyperband and MFMS-GP. We suspect the method only works when fitting LightGBM models on losses.
>
> Comparison with data mixing law:
> We admit that we have difficulties extracting the exact setup (i.e. how many models trained, and at what scales) for the 1B model experiment in the paper. We also tried to look for its exact implementation in code, but to no avail. Our best guess from their description in Sec. 4.2 is: 40 models trained at 70M, and 20 models trained at 160M, 305M, and 410M scales.
>
> Assuming the interpretation is correct, the corresponding number of evaluations comes out to be ~820, which is far beyond the point MFMS-GP (full-scale) converged for most metrics.
>
>
> >**...scalability and effectiveness of the MFMS-BO method on larger number of domains has not been explored**
>
> Indeed scalability with respect to the number of domains is a key concern. Bayesian optimization approaches are known to work well on the order of 20 dimensions, and we believe the same limitation applies to our case. We note that a high number of domains would likely introduce problems for all optimization methods.
>
> >**...discussion on the computation complexity of the proposed method**
>
> In fact, the computational cost of Bayesian optimization (BO) here is minuscule compared to the cost of training language models (LM). Our BO experiments typically take under 10 minutes on a laptop, whereas training a 1B model with 4x H100s took 42 hours. We believe the cost of learning a regression model and fitting scaling law for Regmix and DML to be similarly small compared to LM training. All in all, we expect the cost of most methods for identifying promising data mixtures to be orders of magnitude smaller than the cost of LLM trainings.
>
> That said, the computational complexity of our method is the same as standard GP-based Bayesian optimization: O(N^3) where N is the number of data points (trained models).
>
>
> >**out-of-domain extrapolation results… could help to improve the understanding on its OOD performance**
>
> We want to thank the reviewer for highlighting this aspect in which we can improve the experimental setup. We plan to run more experiments to showcase the framework under a different setting, and we will incorporate the suggestion there.

---

> > ### Comment · Reviewer_H7gH · 2025-08-05
> >
> > Thanks for the author's responses and the additional results. I believe the bayesian optimization methods could be an efficient and powerful alternative of the traditional scaling law study. I would suggest the authors to include more results on various set of training data domains and downstream evaluations in the updated manuscript. I will keep my score.

---

### Official Review · Reviewer_6aWf · 2025-07-09

**Clarity:** 4
**Significance:** 3
**Originality:** 3
**Rating:** 5
**Confidence:** 5

**Summary:**

This paper addresses the problem of selecting an optimal training data distribution for large language models (LLMs). Given access to data from multiple sources (e.g., Wikipedia, Books, GitHub, Reddit), the goal is to determine the optimal mixture of these datasets to maximize model performance. To make this problem tractable in practice, the authors adopt a multi-fidelity Bayesian optimization framework.

Specifically, the authors treat the data mixture, model scale (i.e., number of parameters), and number of training steps as input variables to be jointly optimized. This allows them to adaptively balance training cost with the value of information gained about the performance of a full-scale model.

The underlying objective is modeled using a Gaussian Process with a kernel constructed as the product of three RBF kernels, corresponding to the three input dimensions. To capture expected positive correlations between model performance and both model scale and training steps, a linear mean function is employed. For decision-making, the authors use the Expected Improvement over Cost acquisition function to guide the selection of new configurations.

One particularly interesting aspect of this paper is the experimental validation demonstrating that "low-fidelity" samples can effectively predict full-fidelity performance. Specifically, the authors show that smaller models can reliably forecast the outcomes of larger models, and that earlier training steps provide useful insights into the results of later training steps by training a predictor using 472 language model pre-training runs with varying data compositions from the SlimPajama dataset.

**Questions:**

The paper *above* uses a different acquisition function, and you do mention it in the manuscript (I believe [1] corresponds to your reference [54]). However, my question is more about the overall setup. Is the multi-scale aspect of your approach truly different from prior work? I'm not sure I fully understand the distinction.

To me, the creative application and the empirical results presented are already strong enough to justify a solid contribution. That said, I remain unconvinced that the multi-scale component constitutes a novel contribution on its own. A clearer explanation or comparison with prior multi-fidelity or multi-information source frameworks would help strengthen this point. Alternatively, if the differences are minimal, consider avoiding the term framework — or feel free to clarify why my interpretation may be incorrect.

I did browse the github a little bit but can you improve the documentation to offer the simulator stand out as a contribution? I think other researchers and practioners would appreciate to use the simulator as a benchmark as well.

**Ethical Concerns:**

["NO or VERY MINOR ethics concerns only"]

**Final Justification:**

I didn't change my score, I think this paper is an accept. I would bump my score if the authors convinced me that there is a real novelty on the Bayesian Optimization part OR if they present great results on an real application application; I think that would make this paper a solid accept.

In this version, I see many qualities of a paper that can be accepted: it shows a very clever application of Bayesian optimization for data-mixing for LLM training. That's certainty an interesting topic for several practitioners, and the surrogate model created is also of great interest for many researchers. The paper is very easy-to-read and the exposition is pretty good.

**Limitations:**

yes

**Paper Formatting Concerns:**

Nope

**Quality:**

4

**Strengths And Weaknesses:**

* Strengths*

This is a creative and compelling application of Bayesian optimization to select data mixtures for training large language models. The paper is well written, easy to follow, and the experiments are thoughtfully designed — the synthetic benchmark in particular stands out as a strong component of the validation.

* Weaknesses*

That said, the paper would be stronger if the proposed methodology were demonstrated in a real-world application. As it stands, the experiments serve primarily to validate the approach in controlled settings, but do not yet showcase its effectiveness in a practical deployment scenario.

The use of Bayesian optimization in this context is certainly unique and well-motivated. However, it is not entirely clear that the “multi-scale” aspect of the method is novel. Specifically, how does this differ from prior work on multi-information source optimization, such as:

> [1] M. Poloczek, J. Wang, P. I. Frazier. *Multi-Information Source Optimization*. NeurIPS, 2017.

which also handles discrete fidelities in a Bayesian optimization setting?

If the authors are able to clarify this distinction, I believe this paper would represent a solid and meaningful contribution.

---

> ### Author Rebuttal · Authors · 2025-07-31
>
> # General Response
>
> We sincerely thank the reviewers for their thoughtful feedback and careful evaluation of our work. We are grateful that our paper was well received with positive scores (5, 5, 4, 3). Particularly, we are encouraged that the reviewers appreciate the novelty of our approach (6aWf, H7gH, bchD, FwDJ), the importance of the problem tackled in this work (6aWf, H7gH, bchD, FwDJ), and the performance of MFMS-GP (H7gH, bchD, FwDJ).
>
>
> Reviewer bchD raised valid concerns regarding our use of the simulator and existing Bayesian Optimization (BO) methods. In response, we would like to take this opportunity to clarify and highlight what we view as the main contributions of our work:
> - (a) providing the simulator as a testbed for methodological development (echoing Reviewer 6aWf’s comment)
> - (b) proposing a sequential decision making framework that naturally describes data mixture optimization.
>
> (a) A clear bottleneck of developing methods for data mixture optimization is the high cost of training new LLMs each time a method proposes a different mixture. To make progress, the scientific community needs a high-fidelity but cheap setting in order to develop effective methodologies. Therefore, we decided to build a novel high-fidelity simulator suitable for testing and benchmarking principled algorithms for data mixing. We believe this simulator will be a useful resource for the community, supporting future advancements in this area. We would also like to note that the use of a simulator is a common approach in the study of hyper-parameter optimization where evaluation is expensive [1, 2].
>
> (b) Our motivation for proposing the sequential decision framework stems from the current status-quo in the literature: the existing approaches to data mixing optimization rely on (i) ad-hoc data curation approaches and (ii) standard scaling laws that perform deterministic extrapolation. While using BO for the sequential decision framework may seem natural, its actual effectiveness has not been established in prior work, leaving practitioners either unaware of its potential or hesitant to adopt it. We are among the first to rigorously demonstrate the effectiveness of BO for data-mixing optimization, and this verification is far from trivial—it required training 472 models up to 1B parameters. We hope these promising results will spur further exploration of BO for data mixing and other pretraining optimizations, potentially leading to significant savings in both cost and compute.
>
>
> We hope to address your questions and concerns satisfactorily this round, and are truly thankful for your feedback, which has helped improve the paper in various ways.
>
> [1] Zela et al., Surrogate NAS Benchmarks: Going Beyond the Limited Search Spaces of Tabular NAS Benchmarks, International Conference on Learning Representations, 2022
>
> [2] Eggensperger et al., Efficient benchmarking of hyperparameter optimizers via surrogates, Proceedings of the Twentynineth National Conference on Artificial Intelligence, 2015
>
> # Point-by-Point Responses to Reviewer 6aWf
> We want to thank the reviewer for your encouraging feedback and strong support of our paper. We are especially glad that you found our demonstration of the use of smaller models and earlier training steps interesting.
>
> >**how does this differ from prior work on multi-information source optimization… if the differences are minimal, consider avoiding the term framework**
>
> In making this point, the reviewer points to a paper discussing Multi-Information Source optimization, dealing with a situation in which multiple sources inform the objective function of interest but where each of those sources may provide biased estimates of the true function.
>
> We agree with the reviewer that the idea of having multiple sources informing the true value of interest is not new for Bayesian optimization. At this level of abstraction, the idea indeed encompasses our approach. However, we believe the problem of data mixture optimization introduces a unique and interesting structure that deserves emphasis.
>
> Specifically, there are two key dimensions in which fidelities are introduced: model scale and training steps. The two dimensions are different in a few ways (e.g. significantly more training steps than commonly considered model scales). But most notably, querying an information source at a later training step also yields outputs from all earlier steps as well. However, this feature is not present in the model scale dimension. This asymmetry poses an interesting structure that can potentially be better exploited by developing tailored acquisition functions. We believe this can be an exciting direction for future works.
>
> That said, we are happy to adopt more suitable terminology to reflect better the relationship to prior works while also highlighting the interesting distinctions. We would welcome any suggestions the reviewer may have on alternative phrasing.
>
> >**... offer the simulator stand out as a contribution**
>
> We thank the reviewer for this suggestion. We also think the simulator is one of the key contributions, and will be sure to highlight it in the codebase for the camera-ready version.

---

> > ### Comment · Reviewer_6aWf · 2025-08-03
> >
> > Thanks for your response. I'm quite happy with the answer because it focuses on the paper's main novelty, which is BO for data-mixing optimization. Indeed, I changed my mind about my hesitation around the word "framework." It is helpful to call it a framework when focusing on the probabilistic extrapolation aspect of the data-mixing optimization problem.
> >
> > As a reviewer who has read several Bayesian optimization papers, my bias is to evaluate the novelty/contribution on the optimization side. Still, my reading of this paper is that the application is the real novelty, which, again, I enjoyed, and I agree that future work could further develop it. I'm looking forward to seeing this acquisition function.
> >
> > Putting my bias aside, here are some things that confused me:
> >
> > - The paragraph starting on line 73 is great for me. You formulate the data-mixing optimization as a Bayesian Optimization problem. Perhaps, consider adding multi-fidelity Bayesian optimization? Anyhow, I don't see problems with this paragraph. But the next one says "traditional BO", which immediately reads to me that you are about to suggest a new Bayesian optimization formulation, hence my initial question: why is this different from previous BO formulations?
> >
> > - In the list of contributions, you list "Multi-Fidelity Multi-Scale BO problem" as a novelty. Can you rephrase this? The novelty is the application to data mixture optimization, and, naturally, the application has its features/challenges. The overall Bayesian optimization setting is not novel per se.

---

> ### Author Response · Authors · 2025-08-03
> **Thank you!**
>
> We wholeheartedly agree with all the points the reviewer raised.
> (1) Indeed the novelty lies in the application, and we will adjust the phrasing when listing contributions accordingly.
> (2) We agree that the phrase “traditional BO” is misleading. Our initial hope was that we could prevent misunderstanding by mentioning multi-fidelity BO in the next paragraph (starting on line 83), but we recognize that it is not optimal. We will revise our exposition in those sections.
>
> For the camera-ready version, we will use the extra page available to more carefully discuss our work in relation to the multi-fidelity BO literature. Thank you for pointing out where our exposition could improve. The discussions brought up by the reviewer are extremely helpful for the revision.

---

### Decision · Program_Chairs · 2025-09-17

**Decision:**

Accept (poster)

**Comment:**

This paper presents a novel algorithm for constructing optimal data distributions for LLM training. The authors frame the problem of data distribution selection as a multi-fidelity, multi-scale Bayesian optimization, jointly optimizing data mixtures, model scale, and training steps in an adaptive manner. The proposed method is extensively evaluated at scale, demonstrating substantial empirical gains over random search baselines.

Most reviewers were impressed by the novelty, viewing the framework as an interesting and impactful application of Bayesian optimization.

While several reviewers raised concerns about the lack of real-world deployments and the focus on controlled experimental settings, the strengths of the work outweigh these limitations. There were also concerns about weak baselines, but I find these were adequately addressed during the rebuttal.

Overall, I align with the positive reviews regarding the novelty of the contribution, and I recommend acceptance.